

# Optimally selecting the top $k$ values from $X + Y$ with layer-ordered heaps

Oliver Serang

Department of Computer Science, University of Montana, Missoula, Montana, United States

## ABSTRACT

Selection and sorting the Cartesian sum, $X + Y$, are classic and important problems. Here, a new algorithm is presented, which generates the top $k$ values of the form $X_i + Y_j$. The algorithm relies on layer-ordered heaps, partial orderings of exponentially sized layers. The algorithm relies only on median-of-medians and is simple to implement. Furthermore, it uses data structures contiguous in memory, cache efficient, and fast in practice. The presented algorithm is demonstrated to be theoretically optimal.

## INTRODUCTION

Given two vectors of length $n$, $X$ and $Y$, top-$k$ on $X + Y$ finds the $k$ smallest values of the form $X_i + Y_j$. Note that this problem definition is presented w.l.o.g.; $X$ and $Y$ need not share the same length. Top-$k$ is important to practical applications, such as selecting the most abundant $k$ isotopologue peaks from a compound (*Kreitzberg et al., 2020*). Top-$k$ is $\in \Omega(n + k)$, because loading the vectors is $\in \Theta(n)$ and returning the minimal $k$ values is $\in \Theta(k)$.

### Naive approach

Top-$k$ can be solved trivially in $O(n^2 \log(n) + k) = O(n^2 \log(n))$ steps by generating and sorting all $n^2$ values of the form $X_i + Y_j$. By using median-of-medians (*Blum et al., 1973*), this can be improved to $O(n^2)$ steps by generating all $n^2$ values and performing $k$-selection on them.

### Existing, tree-based methods for top-$k$

In 1982, Frederickson & Johnson introduced a method reminiscent of median-of-medians (*Blum et al., 1973*); their method selects only the $k^{\text{th}}$ minimum value from $X + Y$ in $O\left(n + \min(n, k) \log\left(\frac{k}{\min(n,k)}\right)\right)$ steps (*Frederickson & Johnson, 1982*).

Frederickson subsequently published a second algorithm, which finds the $k$ smallest elements from a min-heap in $O(k)$, assuming the heap has already been built (*Frederickson, 1993*). Combining this method with a combinatoric heap on $X + Y$ (described below for the Kaplan et al. method) solves top-$k$ in $O(n + k)$. The tree data structure in Frederickson's method can be combined with a combinatoric heap to compute the $k^{\text{th}}$ smallest value from $X + Y$.

Corresponding author
Oliver Serang,
oliver.serang@umt.edu

Kaplan et al. described an alternative method for selecting the $k^{th}$ smallest value (*Kaplan et al., 2019*); that method explicitly used Chazelle's soft heaps (*Chazelle, 2000*). By heapifying $X$ and $Y$ in linear time (i.e., guaranteeing w.l.o.g. that $X_i \leq X_{2i}, X_{2i+1}$), $\min_{i,j} X_i + Y_j = X_1 + Y_1$. Likewise, $X_i + Y_j \leq X_{2i} + Y_j, X_{2i+1} + Y_j, X_i + Y_{2j}, X_i + Y_{2j+1}$. The soft heap is initialized to contain tuple $(X_1 + Y_1, 1, 1)$. Then, as tuple $(v, i, j)$ is popped from soft heap, lower-quality tuples are inserted into the soft heap. These lower-quality tuples of $(i,j)$ are

$$\begin{cases} \{(2i, 1), (2i + 1, 1), (i, 2), (i, 3)\}, & j = 1 \\ \{(i, 2j), (i, 2j + 1)\}, & j > 1. \end{cases} \tag{1}$$

In the matrix $X_i + Y_j$ (which is not realized), this scheme progresses in row-major order, thereby avoiding a tuple being added multiple times.

To compute the $k^{th}$ smallest value from $X + Y$, the best $k$ values are popped from the soft heap. Even though only the minimal $k$ values are desired, "corruption" in the soft heap means that the soft heap will not always pop the minimal value; however, as a result, soft heaps can run faster than the $\Omega(n \log(n))$ lower bound on comparison sorting. The free parameter $\varepsilon \in (0, \frac{1}{2}]$ bounds the number of corrupt elements in the soft heap (which may be promoted earlier in the queue than they should be) as $\leq t \cdot \varepsilon$, where $t$ is the number of insertions into the soft heap thus far. Thus, instead of popping $k$ items (and inserting their lower-quality dependents as described in Eq. (1)), the total number of pops, $p$, can be found: The maximal size of the soft heap after $p$ pops is $\leq 3p$ (because each pop removes one element and inserts $\leq 4$ elements according to Eq. (1)); therefore, $p - corruption \geq p - 4p \cdot \varepsilon$, and thus $p - 4p \cdot \varepsilon \geq k$ guarantees that $p - corruption \geq k$. This leads to $p = \frac{k}{1-4\varepsilon}$, $\varepsilon \leq \frac{1}{4}$. This guarantees that $\Theta(k)$ values, which must include the minimal $k$ values, are popped. These values are post-processed to retrieve the minimal $k$ values via linear time one-dimensional selection (*Blum et al., 1973*). For constant $\varepsilon$, both pop and insertion operations to the soft heap are $\in \tilde{O}(1)$, and thus the overall runtime of the algorithm is $\in O(n + k)$.

Note that the Kaplan et al. method easily solves top-$k$ in $O(n + k)$ steps; this is because computing the $k^{th}$ smallest value from $X + Y$ pops the minimal $k$ values from the soft heap.

## Layer-ordered heaps and a novel selection algorithm on X + Y

This paper uses layer-ordered heaps (LOHs) (*Kreitzberg et al., 2020*) to produce an optimal selection algorithm on $X + Y$. LOHs are stricter than heaps but not as strict as sorting: Heaps guarantee only that $X_i \leq X_{children(i)}$, but do not guarantee any ordering between one child of $X_i$, $a$, and the child of the sibling of $a$. Sorting is stricter still, but sorting $n$ values cannot be done faster than $\log_2(n!) \in \Omega(n \log(n))$. LOHs partition the array into several layers such that the values in a layer are $\leq$ to the values in subsequent layers: $X^{(u)} = X^{(u)}_1, X^{(u)}_2, \ldots \leq X^{(u+1)}$. The size of these layers starts with $|X^{(1)}| = 1$ and grows exponentially such that $\lim_{u \to \infty} \frac{|X^{(u+1)}|}{|X^{(u)}|} = \alpha \geq 1$ (note that $\alpha = 1$ is equivalent to sorting because all layers have size 1). By assigning values in layer $u$ children from layer $u + 1$, this can be seen as a more constrained form of the heap; however, unlike sorting, for any constant $\alpha > 1$, LOHs can be constructed $\in O(n)$ by performing iterative linear time

one-dimensional selection, iteratively selecting and removing the largest layer until all layers have been partitioned. For example, 8,1,6,4,5,3,2 can be LOHified with $\alpha = 2$ into an LOH with three layers ($1 \leq 3,2 \leq 8,4,6,5$) by first selecting the largest 4 values on the entire list (8,4,6,5), removing them, and then selecting the largest 2 values from the remaining 3 values (3,2).

Although selections reminiscent of LOHs may have been used previously, formalization of rank $\alpha$ LOHs has been necessary to demonstrate that for $1 \ll \alpha \ll 2$, a combination of LOHs and soft heaps allow generating the minimum $k$ values from $X_1 + X_2 + \cdots + X_m$ (where each $X_i$ has length $n$) in $o(n \cdot m + k \cdot m)$ (*Kreitzberg, Lucke & Serang, 2020*). Furthermore, efficiently constructing an LOH of rank $\alpha$ is not trivial when $\alpha \ll 2$; after all, $\alpha \to 1$ results in layers of size $|X^{(1)}| = |X^{(2)}| = \cdots = 1$, indicating a sorting, which implies a runtime $\in \Omega(n\log(n))$ (*Pennington et al., 2020*).

A `python` implementation of a LOH is shown in listing 1.

### Contribution in this manuscript

The new, optimal algorithm for solving top-$k$ presented here makes extensive use of LOHs. It is simple to implement, does not rely on anything more complicated than linear time one-dimensional selection (i.e., it does not use soft heap). Due to its simplicity and contiguous memory access, it has a fast performance in practice.

## METHODS

### Algorithm

The algorithm presented is broken into phases. An illustration of these phases is provided in Fig. 1.

#### Phase 0

The algorithm first LOHifies (i.e., constructs a layer order heap from) both $X$ and $Y$. This is performed by using linear time one-dimensional selection to iteratively remove the largest remaining layer (i.e., the simplest LOH construction method, which is optimal when $\alpha \gg 1$).

#### Phase 1

Now layer products of the form

$X^{(u)} + Y^{(v)} = X_1^{(u)} + Y_1^{(v)}, X_1^{(u)} + Y_2^{(v)}, \ldots, X_2^{(u)} + Y_1^{(v)}, \ldots$ are considered, where $X^{(u)}$ and $Y^{(v)}$ are layers of their respective LOHs.

In phases 1–2, the algorithm initially considers only the minimum and maximum values in each layer product: $\lfloor (u,v) \rfloor = (\min(X^{(u)} + Y^{(v)}), (u,v), 0)$, $\lceil (u,v) \rceil = (\max(X^{(u)} + Y^{(v)}), (u,v), 1)$. It is unnecessary to compute the Cartesian product of values to build a layer product; instead, only the minimum or maximum values in $X^{(u)}$ and $Y^{(v)}$ are needed. Note that the final value in the tuple uses 0 to indicate that this is the minimum value in the layer product or 1 to indicate that this is the maximum value in the layer product; this ensures that even layer products with homogeneous values satisfy $\lfloor (u,v) \rfloor < \lceil (u,v) \rceil$. Scalar values can be compared to tuples: $X_i + Y_j \leq \lceil (u,v) \rceil = (\max(X^{(u)} + Y^{(v)}), (u,v), 1) \leftrightarrow X_i + Y_j \leq \max(X^{(u)} + Y^{(v)})$.

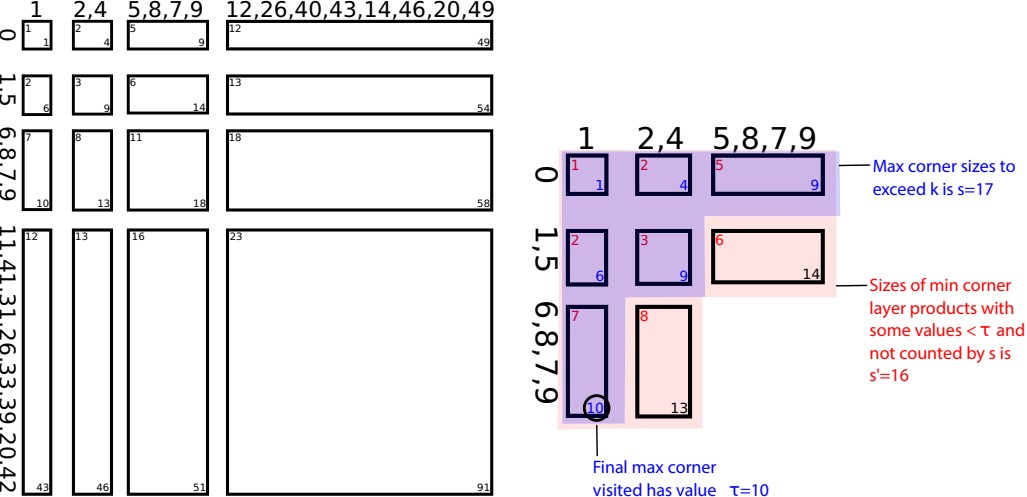

**Figure 1 Illustration of method for selecting the $k = 14$ minimal values from $X + Y$: Phase 0: $X = \{31,5,11,7,33,6,39,42,20,0,9,1,41,26,8\}$ and $Y = \{12,26,40,9,14,49,8,2,20,1,46,43,4,5,7\}$ are both LOHified to axes in $O(n)$ time.** Note that the minimum and maximum values in a layer are placed at the first and last positions in the layer, respectively; otherwise values within layers are themselves unordered. Phase 1: The minimum and maximum corners of all layer products (grid) are visited together in ascending order until the area of the layer products whose max corners are visited exceeds $k$ (inset), and the largest value visited is labeled as $\tau = 10$. Phase 2: The layer products whose max corners have been visited (blue) has area $s$ that exceeds $k$ but has $s \in O(k)$. Likewise, the layer products whose min corners have been visited but whose max corners have not been visited, and which therefore contain some elements $< \tau$, have area $s' \in O(k)$. Phase 3: Together, these layer products (red and blue) contain all values that may be in minimal $k = 14$. Since there are $O(k)$ such values, they can be selected using median-of-medians in $O(k)$ time.

Binary heap $H$ is initialized to contain tuple $\lfloor (1,1) \rfloor$. A set of all tuples in $H$ is maintained to prevent duplicates from being inserted into $H$ (this set could be excluded by using the Kaplan et al. proposal scheme). The algorithm proceeds by popping the lexicographically minimum tuple from $H$. W.l.o.g., there is no guaranteed ordering of the form $X^{(u)} + Y^{(v)} \leq X^{(u+1)} + Y^{(v)}$, because it may be that $\max(X^{(u)} + Y^{(v)}) > \min(X^{(u+1)} + Y^{(v)})$; however, lexicographically, $\lfloor (u,v) \rfloor < \lfloor (u+1,v) \rfloor, \lfloor (u,v+1) \rfloor, \lceil (u,v) \rceil$; thus, the latter tuples need be inserted into $H$ only after $\lfloor (u,v) \rfloor$ has been popped from $H$. Note that for this reason and to break ties where layer products contain identical values, $(u,v)$ are included in the tuple. $\lceil (u,v) \rceil$ tuples do not insert any new tuples into $H$ when they're popped.

Whenever a tuple of the form $\lceil (u,v) \rceil$ is popped from $H$, the index $(u,v)$ is appended to list $q$ and the size of the layer product $|X^{(u)} + Y^{(v)}| = |X^{(u)}| \cdot |Y^{(v)}|$ is accumulated into integer $s$. This method proceeds until $s \geq k$.

### Phase 2

Any remaining tuple in $H$ of the form $\left( \max(X^{(u')} + Y^{(v')}), (u',v'), 1 \right)$ has its index $(u',v')$ appended to list $q$. $s'$ is the total number of elements in each of these $(u',v')$ layer products appended to $q$ during phase 2.

*Phase 3*

The values from every element in each layer product in $q$ are generated. A linear time one-dimensional $k$-selection is performed on these values and returned.

## Proof of correctness

Lemma 2.4 proves that at termination all layer products found in $q$ must contain the minimal $k$ values in $X + Y$. Thus, by performing one-dimensional $k$-selection on those values in phase 3, the minimal $k$ values in $X + Y$ are found.

**Lemma 2.1.** *If $\lfloor (u, v) \rfloor$ is popped from H, then both $\lfloor (u-1, v) \rfloor$ (if u > 1) and $\lfloor (u, v-1) \rfloor$ (if v > 1) must previously have been popped from H.*

*Proof.* There is a chain of pops and insertions backwards from $\lfloor (u, v) \rfloor$ to $\lfloor (1, 1) \rfloor$.

When both $u, v = 1$, the lemma is true.

W.l.o.g. if $u = 1$ this chain is of the form $\lfloor (1,1) \rfloor, \ldots, \lfloor (1, 2) \rfloor, \ldots, \lfloor (1, 3) \rfloor, \ldots, \lfloor (u, v) \rfloor$, proving the lemma for that case.

Otherwise, both $u, v > 1$. Because insertions into $H$ increment either row or column, something of the form $\lfloor (a, v-1) \rfloor$ with $a \le u$ must be inserted into $H$ before inserting $\lfloor (u, v) \rfloor$. $\lfloor (a, v-1) \rfloor < \lfloor (u, v) \rfloor$, so $\lfloor (a, v-1) \rfloor$ must precede $\lfloor (u, v) \rfloor$ in the chain of pops. If $a = u$, then $\lfloor (u, v-1) \rfloor$ is popped before $\lfloor (u, v) \rfloor$. If $a < u$, then from the insertion of $\lfloor (a, v-1) \rfloor$ into $H$, until $\lfloor (u, v-1) \rfloor$ is popped, $H$ must contain something of the form $\lfloor (a', v-1) \rfloor : \sim a' \le u$, because popping $\lfloor (a', v-1) \rfloor$ inserts $\lfloor (a'+1, v-1) \rfloor$. $\lfloor (a', v-1) \rfloor < \lfloor (u, v) \rfloor$ when $a' \le u$; therefore, $\lfloor (u, v) \rfloor$ cannot be popped before any $\lfloor (a', v-1) \rfloor$ currently in $H$. Because there are a finite number of these $a'$ and they are not revisited, before $\lfloor (u, v) \rfloor$ is popped, $\lfloor (u, v-1) \rfloor$ must be popped. This same process can be repeated with $\lfloor (u-1, b) \rfloor : \sim b \le v$ to show that $\lfloor (u-1, v) \rfloor$ must be popped before $\lfloor (u, v) \rfloor$, proving the lemma for the final case. □

**Lemma 2.2** *If $\lceil (u, v) \rceil$ is popped from H, then both $\lceil (u-1, v) \rceil$ (if u>1) and $\lceil (u, v-1) \rceil$ (if v > 1) must previously have been popped from H.*

*Proof. Inserting $\lceil (u, v) \rceil$ requires previously popping $\lfloor (u, v) \rfloor$. By lemma 2.1, this requires previously popping $\lfloor (u-1, v) \rfloor$ (if u > 1) and $\lfloor (u, v-1) \rfloor$ (if v > 1). These pops will insert $\lceil (u-1, v) \rceil$ and $\lceil (u, v-1) \rceil$ respectively. Thus, $\lceil (u-1, v) \rceil$ and $\lceil (u, v-1) \rceil$, which are both < $\lceil (u, v) \rceil$, are inserted before $\lceil (u, v) \rceil$, and will therefore be popped before $\lceil (u, v) \rceil$.* □

**Lemma 2.3** *All tuples will be visited in ascending order as they are popped from H.*

*Proof.* Let $\lfloor (u, v) \rfloor$ be popped from $H$ and let $\lfloor (a, b) \rfloor < \lfloor (u, v) \rfloor$. Either w.l.o.g. $a < u$, $b \le v$, or w.l.o.g. $a < u$, $b > v$. In the former case, $\lfloor (a, b) \rfloor$ will be popped before $\lfloor (u, v) \rfloor$ by applying induction to lemma 2.1.

In the latter case, lemma 2.1 says that $\lfloor (a, v) \rfloor$ is popped before $\lfloor (u, v) \rfloor$. $\lfloor (a, v) \rfloor < \lfloor (a, v+1) \rfloor < \lfloor (a, v+1) \rfloor < \cdots < \lfloor (a, b) \rfloor < \lfloor (u, v) \rfloor$, meaning that $\forall r \in [v, b], \lfloor (a, r) \rfloor < \lfloor (u, v) \rfloor$. After $\lfloor (a, v) \rfloor$ is inserted (necessarily before it is popped), at least one such $\lfloor (a, r) \rfloor$ must be in $H$ until $\lfloor (a, b) \rfloor$ is popped. Thus, all such $\lfloor (a, r) \rfloor$ will be popped before $\lfloor (u, v) \rfloor$.

Ordering on popping with $\lceil (a, b) \rceil < \lceil (u, v) \rceil$ is shown in the same manner: For $\lceil (u, v) \rceil$ to be in $H$, $\lfloor (u, v) \rfloor$ must have previously been popped. As above, whenever

$\lceil(u, v)\rceil$ is in $H$, then $\lfloor(a, v)\rfloor$ must have been popped, inserting $\lfloor(a, v + 1)\rfloor$ into $H$. Each $\lfloor(a, r)\rfloor$ popped inserts

$\lfloor(a, r + 1)\rfloor$, so at least one $\lfloor(a, r)\rfloor, r \in [v, b]$ must also be in $H$ until $\lfloor(a, b)\rfloor$ is popped. These $\lfloor(a, r)\rfloor \leq \lfloor(a, b)\rfloor < \lceil(a, b)\rceil < \lceil(u, v)\rceil$, and so $\lceil(a, b)\rceil$ will be popped before $\lceil(u, v)\rceil$.

Identical reasoning also shows that $\lfloor(a, b)\rfloor$ will pop before $\lceil(u, v)\rceil$ if $\lfloor(a, b)\rfloor < \lceil(u, v)\rceil$ or if $\lceil(a, b)\rceil < \lfloor(u, v)\rfloor$.

Thus, all tuples are popped in ascending order. □

**Lemma 2.4** *At the end of phase 2, the layer products whose indices are found in q contain the minimal k values.*

*Proof.* Let $(u,v)$ be the layer product that first makes $s \geq k$. There are at least $k$ values of $X + Y$ that are $\leq \max(X^{(u)} + Y^{(v)})$; this means that $\tau = (\text{select}(X + Y, k)) \leq \max(X^{(u)} + Y^{(v)})$. The quality of the elements in layer products in $q$ at the end of phase 1 can only be improved by trading some value for a smaller value, and thus require a new value $< \max(X^{(u)} + Y^{(v)})$.

By lemma 2.3, tuples will be popped from $H$ in ascending order; therefore, any layer product $(u', v')$ containing values $< \max(X^{(u)} + Y^{(v)})$ must have had $\lfloor(u', v')\rfloor$ popped before $\lceil(u, v)\rceil$. If $\lceil(u', v')\rceil$ was also popped, then this layer product is already included in $q$ and cannot improve it. Thus the only layers that need be considered further have had $\lfloor(u', v')\rfloor$ popped but not $\lceil(u', v')\rceil$ popped; these can be found by looking for all $\lceil(u', v')\rceil$ that have been inserted into $H$ but not yet popped.

Phase 2 appends to $q$ all such remaining layer products of interest. Thus, at the end of phase 2, $q$ contains all layer products that will be represented in the $k$-selection of $X + Y$. □

A `python` implementation of this method is shown in listing 2.

## Runtime

Theorem 2.8 proves that the total runtime is $\in O(n + k)$.

**Lemma 2.5** *Let $(u',v')$ be a layer product appended to q during phase 2. Either $u' = 1$, $v' = 1$, or $(u' - 1, v' - 1)$ was already appended to q in phase 1.*

*Proof.* Let $u' > 1$ and $v' > 1$. By lemma 2.3, minimum and maximum layer products are popped in ascending order. By the layer ordering property of $X$ and $Y$, $\max(X^{(u'-1)}) \leq \min(X^{(u')})$ and $\max(Y^{(v'-1)}) \leq \min(Y^{(v')})$. Thus, $\lceil(u' - 1, v' - 1)\rceil < \lfloor(u', v')\rfloor$ and so $\lceil(u' - 1, v' - 1)\rceil$ must be popped before $\lfloor(u', v')\rfloor$. □

**Lemma 2.6** *s, the number of elements in all layer products appended to q in phase 1, is $\in O(k)$.*

*Proof.* $(u,v)$ is the layer product whose inclusion during phase 1 in $q$ achieves $s \geq k$; therefore, $s - |X^{(u)} + Y^{(v)}| < k$. This happens when $\lceil(u, v)\rceil$ is popped from $H$.

*If $k = 1$, popping $\lceil(1, 1)\rceil$ ends phase 1 with $s = 1 \in O(k)$.*

If $k > 1$, then at least one layer index is $> 1$: $u > 1$ or $v > 1$. W.l.o.g., let $u > 1$. By lemma 2.1, popping $\lceil(u, v)\rceil$ from $H$ requires previously popping $\lceil(u - 1, v)\rceil$. $|X^{(u)} + Y^{(v)}| = |X^{(u)}| \cdot |Y^{(v)}| \approx \alpha \cdot |X^{(u-1)}| \cdot |Y^{(v)}| = \alpha \cdot |X^{(u-1)} + Y^{(v)}|$ (where $\approx$ indicates asymptotic behavior); therefore, $|X^{(u)} + Y^{(v)}| \in O(|X^{(u-1)} + Y^{(v)}|)$. $|X^{(u-1)} + Y^{(v)}|$ is already counted in $s - |X^{(u)} +$

$Y^{(v)}| < k$, and so $|X^{(u-1)} + Y^{(v)}| < k$ and $|X^{(u)} + Y^{(v)}| \in O(k)$. $s < k + |X^{(u)} + Y^{(v)}| \in O(k)$ and hence $s \in O(k)$. □

**Lemma 2.7** $s'$, *the total number of elements in all layer products appended to q in phase 2,* $\in$ *O(k).*

*Proof.* Each layer product appended to $q$ in phase 2 has had $\lfloor(u', v')\rfloor$ popped in phase 1. By lemma 2.5, either $u' = 1$ or $v' = 1$ or $\lceil(u' - 1, v' - 1)\rceil$ must have been popped before $\lfloor(u', v')\rfloor$.

First consider when $u' > 1$ and $v' > 1$. Each $(u',v')$ matches exactly one layer product $(u' - 1,v' - 1)$. Because $\lceil(u' - 1, v' - 1)\rceil$ must have been popped before $\lfloor(u', v')\rfloor$, then $\lceil(u' - 1, v' - 1)\rceil$ was also popped during phase 1. $s$, the count of all elements whose layer products were inserted into $q$ in phase 1, includes $|X^{(u'-1)} + Y^{(v'-1)}|$ but does not include $X^{(u')} + Y^{(v')}$ (the latter is appended to $q$ during phase 2). By exponential growth of layers in $X$ and $Y$, $|X^{(u')} + Y^{(v')}| \approx \alpha^2 \cdot |X^{(u'-1)} + Y^{(v'-1)}|$. These $|X^{(u' - 1)} + Y^{(v' - 1)}|$ values were included in $s$ during phase 1, and thus the total number of elements in all such $(u' - 1,v' - 1)$ layer products is $\leq s$. Thus the sum of sizes of all layer products $(u',v')$ with $u' > 1$ and $v' > 1$ that are appended to $q$ during phase 2 is asymptotically $\leq \alpha^2 \cdot s$.

When either $u' = 1$ or $v' = 1$, the number of elements in all layer products must be $\in O$ $(n)$: $\sum_{u'} |X^{(u')} + Y^{(1)}| + \sum_{v'} |X^{(u')} + Y^{(1)}| < 2n$; however, it is possible to show that contributions where $u' = 1$ or $v' = 1$ are $\in O(k)$:

W.l.o.g. for $u'>1$, $\lfloor(u', 1)\rfloor$ is inserted into $H$ only when $\lfloor(u'-1, 1)\rfloor$ is popped. Thus at most one $\lfloor(u', 1)\rfloor$ can exist in $H$ at any time. Furthermore, popping $\lfloor(u', 1)\rfloor$ from $H$ requires previously popping $\lceil(u'-1, 1)\rceil$ from $H$: layer ordering on $X$ implies max $(X^{(u' - 1)}) \leq \min(X^{(u')})$ and $|Y^{(1)}| = 1$ implies $\min(Y^{(1)}) = \max(Y^{(1)})$, and so $\lceil(u' - 1, 1)\rceil = (\max(X^{(u'-1)} + Y^{(1)}), (u' - 1, 1), 1) < \lfloor(u', 1)\rfloor = (\min(X^{(u')} + Y^{(1)}), (u', 1), 0)$. Thus $\lceil(u'-1, 1)\rceil$ has been popped from $H$ and counted in $s$. By the exponential growth of layers, the contribution of all such $u' > 1, v' = 1$ will be $\approx \leq \alpha \cdot s$, and so the contributions of $u' > 1$, $v' = 1$ or $u' = 1, v' > 1$ will be $\approx \leq 2\alpha \cdot s$.

When $u' = v' = 1$, the layer product contains 1 element.

Therefore, $s'$, the total number of elements found in layer products appended to $q$ during phase 2, has $s' \leq (\alpha^2 + 2\alpha) \cdot s + 1$. By lemma 2.6, $s \in O(k)$, and thus $s' \in O(k)$. □

**Theorem 2.8** *The total runtime of the algorithm is* $\in$ *O(n + k).*

*Proof.* For any constant $\alpha > 1$, LOHification of $X$ and $Y$ runs in linear time, and so phase 0 runs $\in O(n)$.

The total number of layers in each LOH is $\approx \log_\alpha(n)$; therefore, the total number of layer products is $\approx \log_\alpha^2(n)$. In the worst-case scenario, the heap insertions and pops (and corresponding set insertions and removals) will sort $\approx 2 \log_\alpha^2(n)$ elements, because each layer product may be inserted as both $\lfloor \cdot \rfloor$ *or* $\lceil \cdot \rceil$; the worst-case runtime via comparison sort will be $\in O(\log_\alpha^2(n) \log(\log_\alpha^2(n))) \subset o(n)$. The operations to maintain a set of indices in the heap have the same runtime per operation as those inserting/removing to a binary heap, and so can be amortized out. Thus, the runtimes of phases 1–2 are amortized out by the $O(n)$ runtime of phase 0.

**Table 1 Average runtimes (in seconds) on random uniform integer $X$ and $Y$ with $|X| = |Y| = n$.** The layer-ordered heap implementation used $\alpha = 2$ and resulted in $s + s'/k = 3.637$ on average. Individual and total runtimes are rounded to three significant figures.

| | Naive $n^2\log(n) + k$ | Kaplan et al. soft heap | Layer-ordered heap (total = phase 0 + phases 1–3) |
|---|---|---|---|
| $n = 1{,}000, k = 250$ | 0.939 | 0.0511 | 0.00892 = 0.00693 + 0.002 |
| $n = 1{,}000, k = 500$ | 0.952 | 0.099 | 0.0102 = 0.00648 + 0.00374 |
| $n = 1{,}000, k = 1{,}000$ | 0.973 | 0.201 | 0.014 = 0.00764 + 0.00639 |
| $n = 1{,}000, k = 2{,}000$ | 0.953 | 0.426 | 0.0212 = 0.00652 + 0.0146 |
| $n = 1{,}000, k = 4{,}000$ | 0.950 | 0.922 | 0.0278 = 0.00713 + 0.0206 |
| $n = 2{,}000, k = 500$ | 4.31 | 0.104 | 0.0194 = 0.0160 + 0.00342 |
| $n = 2{,}000, k = 1{,}000$ | 4.11 | 0.203 | 0.0211 = 0.0139 + 0.00728 |
| $n = 2{,}000, k = 2{,}000$ | 4.17 | 0.432 | 0.0254 = 0.0140 + 0.0114 |
| $n = 2{,}000, k = 4{,}000$ | 4.16 | 0.916 | 0.0427 = 0.0147 + 0.0280 |
| $n = 2{,}000, k = 8{,}000$ | 4.13 | 2.03 | 0.0761 = 0.0143 + 0.0617 |
| $n = 4{,}000, k = 1{,}000$ | 17.2 | 0.207 | 0.0507 = 0.0459 + 0.00488 |
| $n = 4{,}000, k = 2{,}000$ | 17.2 | 0.422 | 0.409 = 0.0268 + 0.0141 |
| $n = 4{,}000, k = 4{,}000$ | 17.1 | 0.907 | 0.0481 = 0.0277 + 0.0205 |
| $n = 4{,}000, k = 8{,}000$ | 17.3 | 1.98 | 0.0907 = 0.0278 + 0.0629 |
| $n = 4{,}000, k = 16{,}000$ | 17.3 | 4.16 | 0.133 = 0.0305 + 0.103 |

Lemma 2.6 shows that $s \in O(k)$. Likewise, lemma 2.7 shows that $s' \in O(k)$. The number of elements in all layer products in $q$ during phase 3 is $s + s' \in O(k)$. Thus, the number of elements on which the one-dimensional selection is performed will be $\in O(k)$. Using a linear time one-dimensional selection algorithm, the runtime of the $k$-selection in phase 3 is $\in O(k)$.

The total runtime of all phases $\in O(n + k + k + k) = O(n + k)$. $\square$

## RESULTS

Runtimes of the naive $O(n^2\log(n) + k)$ method (chosen for reference because it is the easiest method to implement and because of the fast runtime constant on python's built-in sorting routine), the soft heap-based method from Kaplan et al., and the LOH-based method in this paper are shown in Table 1. The proposed approach achieves a >295× speedup over the naive approach and >18× speedup over the soft heap approach. LOHs are more lightweight than soft heaps, including contiguous memory access patterns and far fewer pointer dereferences than soft heaps.

## DISCUSSION

The algorithm can be thought of as "zooming out" as it pans through the layer products, thereby passing the value threshold at which the $k^{\text{th}}$ best value $X_i + Y_j$ occurs. It is somewhat reminiscent of skip lists (*Pugh, 1990*); however, where a skip list begins coarse and progressively refines the search, this approach begins finely and becomes progressively coarser. The notion of retrieving the best $k$ values while "overshooting" the target by as little as possible results in some values that may be considered but which will

not survive the final one-dimensional selection in phase 3. This is reminiscent of "corruption" in Chazelle's soft heaps. Like soft heaps, this method eschews sorting in order to prevent a runtime $\in \Omega(n\log(n))$ or $\in \omega(k\log(k))$. But unlike soft heaps, LOHs can be constructed easily using only an implementation of median-of-medians (or any other linear time one-dimensional selection algorithm).

Phase 3 is the only part of the algorithm in which $k$ appears in the runtime formula. This is significant because the layer products in $q$ at the end of phase 2 could be returned in their compressed form (i.e., as the two layers to be combined). The total runtime of phases 0–2 is $\in O(n)$. It may be possible to recursively perform $X + Y$ selection on layer products $X^{(u)} + Y^{(v)}$ to compute layer products constituting exactly the $k$ values in the solution, still in factored Cartesian layer product form. Similarly, it may be possible to perform the one-dimensional selection without fully inflating every layer product into its constituent elements. For some applications, a compressed form may be acceptable, thereby making it plausible to remove the requirement that the runtime be $\in \omega(k)$.

As noted in theorem 2.8, even fully sorting all of the minimum and maximum layer products would be $\in o(n)$; sorting in this manner may be preferred in practice, because it simplifies the implementation (Listing 3) at the cost of incurring greater runtime in practice when $k \ll n^2$. Furthermore, listing 3 is unsuitable for online processing (i.e., where $X$ and $Y$ are extended on the fly or where several subsequent selections are performed), whereas listing 2 could be adapted to those uses.

Phase 0 (which performs LOHification) is the slowest part of the presented python implementation; it would benefit from having a practically faster implementation to perform LOHify.

The fast practical performance is partially due to the algorithm's simplicity and partially due to the contiguous nature of LOHs. Online data structures like soft heap are less easily suited to contiguous access, because they support efficient removal and therefore move pointers to memory rather than moving the contents of the memory.

The choice of $\alpha$ affects performance through the cost of LOHifying and the amount by which the number of generated values overshoots the $k$ minimum values wanted: when $\alpha \approx 1$, LOHify effectively sorts $X$ and $Y$, but generates few extra values; $\alpha \gg 1$, LOHify has a linear runtime, but generates more extra values, which need to be removed by the final $k$-selection.

## CONCLUSION

LOHs can be constructed in linear time and used to produce a theoretically optimal algorithm for selecting the minimal $k$ values from $X + Y$. The new optimal algorithm presented here is faster in practice than the existing soft heap-based optimal algorithm.

## APPENDIX

### Python code

**Listing 1.** `LayerOrderedHeap.py`: A class for LOHifying, retrieving layers, and the minimum and maximum value in a layer.

```
# https://stackoverflow.com/questions/10806303/python-implementation-of-median-of-
medians-algorithm
def median_of_medians_select(L, j): # returns j-th smallest value:
    if len(L) < 10:
        L.sort()
        return L[j]
    S = []
    lIndex = 0
    while lIndex+5 < len(L)-1:
        S.append(L[lIndex:lIndex+5])
        lIndex += 5
    S.append(L[lIndex:])
    Meds = []
    for subList in S:
        Meds.append(median_of_medians_select(subList, int((len(subList)-1)/2)))
    med = median_of_medians_select(Meds, int((len(Meds)-1)/2))
    L1 = []
    L2 = []
    L3 = []
    for i in L:
        if i < med:
            L1.append(i)
        elif i > med:
            L3.append(i)
        else:
            L2.append(i)
    if j < len(L1):
        return median_of_medians_select(L1, j)
    elif j < len(L2) + len(L1):
        return L2[0]
    else:
        return median_of_medians_select(L3, j-len(L1)-len(L2))

def partition(array, left_n):
    n = len(array)
    right_n = n - left_n

    # median_of_medians_select argument is index, not size:
    max_value_in_left = median_of_medians_select(array, left_n-1)

    left = []
```

```
        right = []
        for i in range(n):
            if array[i] < max_value_in_left:
                left.append(array[i])
            elif array[i] > max_value_in_left:
                right.append(array[i])
        num_at_threshold_in_left = left_n - len(left)
        left.extend([max_value_in_left]*num_at_threshold_in_left)
        num_at_threshold_in_right = right_n - len(right)
        right.extend([max_value_in_left]*num_at_threshold_in_right)
        return left, right

def layer_order_heapify_alpha_eq_2(array):
    n = len(array)
    if n == 0:
        return []
    if n == 1:
        return array
    new_layer_size = 1
    layer_sizes = []
    remaining_n = n
    while remaining_n > 0:
        if remaining_n >= new_layer_size:
            layer_sizes.append(new_layer_size)
        else:
            layer_sizes.append(remaining_n)
        remaining_n -= new_layer_size
        new_layer_size *= 2
    result = []
    for i,ls in enumerate(layer_sizes[::-1]):
        small_vals,large_vals = partition(array, len(array) - ls)
        array = small_vals
        result.append(large_vals)
    return result[::-1]

class LayerOrderedHeap:
    def __init__(self, array):
        self._layers = layer_order_heapify_alpha_eq_2(array)
        self._min_in_layers = [ min(layer) for layer in self._layers ]
        self._max_in_layers = [ max(layer) for layer in self._layers ]
        #self._verify()

    def __len__(self):
```

```
        return len(self._layers)

    def _verify(self):
        for i in range(len(self)-1):
            assert(self.max(i) <= self.min(i+1))

    def __getitem__(self, layer_num):
        return self._layers[layer_num]

    def min(self, layer_num):
        assert( layer_num < len(self) )
        return self._min_in_layers[layer_num]

    def max(self, layer_num):
        assert( layer_num < len(self) )
        return self._max_in_layers[layer_num]

    def __str__(self):
        return str(self._layers)
```

**Listing 2.** `CartesianSumSelection.py`: A class for efficiently performing selection on $X + Y$ in $\Theta(n + k)$ steps.

```
from LayerOrderedHeap import *
import heapq

class CartesianSumSelection:
    def _min_tuple(self,i,j):
        # True for min corner, False for max corner
        return (self._loh_a.min(i) + self._loh_b.min(j), (i,j), False)

    def _max_tuple(self,i,j):
        # True for min corner, False for max corner
        return (self._loh_a.max(i) + self._loh_b.max(j), (i,j), True)

    def _in_bounds(self,i,j):
        return i < len(self._loh_a) and j < len(self._loh_b)

    def _insert_min_if_in_bounds(self,i,j):
        if not self._in_bounds(i,j):
            return
```

```python
        if (i,j,False) not in self._hull_set:
            heapq.heappush(self._hull_heap, self._min_tuple(i,j))
            self._hull_set.add( (i,j,False) )

    def _insert_max_if_in_bounds(self,i,j):
        if not self._in_bounds(i,j):
            return

        if (i,j,True) not in self._hull_set:
            heapq.heappush(self._hull_heap, self._max_tuple(i,j))
            self._hull_set.add( (i,j,True) )

    def __init__(self, array_a, array_b):
        self._loh_a = LayerOrderedHeap(array_a)
        self._loh_b = LayerOrderedHeap(array_b)
        self._hull_heap = [ self._min_tuple(0,0) ]
        # False for min:
        self._hull_set = { (0,0,False) }

        self._num_elements_popped = 0
        self._layer_products_considered = []

        self._full_cartesian_product_size = len(array_a) * len(array_b)

    def _pop_next_layer_product(self):
        result = heapq.heappop(self._hull_heap)
        val, (i,j), is_max = result
        self._hull_set.remove( (i,j,is_max) )

        if not is_max:
            # when min corner is popped, push their own max and neighboring mins
            self._insert_min_if_in_bounds(i+1,j)
            self._insert_min_if_in_bounds(i,j+1)
            self._insert_max_if_in_bounds(i,j)
        else:
            # when max corner is popped, do not push
            self._num_elements_popped += len(self._loh_a[i]) * len(self._loh_b[j])
            self._layer_products_considered.append( (i,j) )

    return result
    def select(self, k):
        assert( k <= self._full_cartesian_product_size )

    while self._num_elements_popped < k:
```

```
            self._pop_next_layer_product()

        # also consider all layer products still in hull
        for val, (i,j), is_max in self._hull_heap:
            if is_max:
                self._num_elements_popped += len(self._loh_a[i]) * len(self._loh_b[j])
                self._layer_products_considered.append( (i,j) )

        # generate: values in layer products

        # Note: this is not always necessary, and could lead to a potentially large speedup.
        candidates = [ val_a+val_b for i,j in self._layer_products_considered for val_a in
        self._loh_a[i] for val_b in self._loh_b[j] ]
        print( 'Ratio of total popped candidates to k: {}'.format(len(candidates) / k) )
        k_small_vals, large_vals = partition(candidates, k)
        return k_small_vals
```

**Listing 3.** `SimplifiedCartesianSumSelection.py`: A simplified implementation of Listing 2. This implementation is slower when $k \ll n^2$; however, it has the same asymptotic runtime for any $n$ and $k$: $\Theta(n + k)$.

```
from LayerOrderedHeap import *

class SimplifiedCartesianSumSelection:
    def _min_tuple(self,i,j):
        # True for min corner, False for max corner
        return (self._loh_a.min(i) + self._loh_b.min(j), (i,j), False)

    def _max_tuple(self,i,j):
        # True for min corner, False for max corner
        return (self._loh_a.max(i) + self._loh_b.max(j), (i,j), True)

    def __init__(self, array_a, array_b):
        self._loh_a = LayerOrderedHeap(array_a)
        self._loh_b = LayerOrderedHeap(array_b)

        self._full_cartesian_product_size = len(array_a) * len(array_b)
        self._sorted_corners = sorted([self._min_tuple(i,j) for i in range(len(self._loh_a)) for
        j in range(len(self._loh_b))] + [self._max_tuple(i,j) for i in range(len(self._loh_a)) for
        j in range(len(self._loh_b))])

    def select(self, k):
        assert( k <= self._full_cartesian_product_size )

        candidates = []
```

```
        index_in_sorted = 0
        num_elements_with_max_corner_popped = 0
        while num_elements_with_max_corner_popped < k:
            val, (i,j), is_max = self._sorted_corners[index_in_sorted]
            new_candidates = [ v_a+v_b for v_a in self._loh_a[i] for v_b in self._loh_b[j] ]
            if is_max:
                num_elements_with_max_corner_popped += len(new_candidates)
            else:
                # Min corners will be popped before corresponding max corner;
                # this gets a superset of what is needed (just as in phase 2)
                candidates.extend(new_candidates)
            index_in_sorted += 1

        print( 'Ratio of total popped candidates to k: {}'.format(len(candidates) / k) )
        k_small_vals, large_vals = partition(candidates, k)
        return k_small_vals
```

## ACKNOWLEDGEMENTS

Thanks to Patrick Kreitzberg, Kyle Lucke, and Jake Pennington for fruitful discussions and kindness.

### Funding

This work was supported by NSF CAREER grant 1845465. The funders had no role in study design, data collection and analysis, decision to publish, or preparation of the manuscript.

### Grant Disclosures

The following grant information was disclosed by the authors:
NSF CAREER: 1845465.

### Competing Interests

The author declares that he has no competing interests.

### Author Contributions

- Oliver Serang conceived and designed the experiments, performed the experiments, analyzed the data, performed the computation work, prepared figures and/or tables, authored or reviewed drafts of the paper, and approved the final draft.

### Data Availability

The data is available at figshare: Serang (2021): Selection on X+Y: python source. figshare. Software. https://doi.org/10.6084/m9.figshare.13708564.v1.

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
