# Peer review of "Optimally selecting the top k values from X + Y with layer-ordered heaps"

_PeerJ Computer Science, doi:10.7717/peerj-cs.501_

## Round 0.1 · original submission · Major Revisions

Please ensure that the lemmas and proofs are rigorously handled.

Reviewer 1 ·

Basic reporting

The authors present a way to find top k smallest elements from the Cartesian sum X+Y where X and Y are two given sets. For this, they use layer-ordered heaps (LOH). In the results section they compare their method with two other methods: 1) naive O(n^2 log(n) + k) method and 2) soft heap based method by Kaplan et. al. They see a 295x and 18x speedup respectively. The authors have also thoroughly analyzed the proof of correctness of the algorithm. Additionally, they have provided the python code for the same.

In my opinion, the paper is well written (there are typos which are listed below) and is organized properly. All the lemmas and theorems are clearly defined. The manuscript also includes enough background on the topics that are needed to understand the paper. But it would have been easily understandable if an example had been provided. Additionally, the space complexity is not analyzed properly.

Typos: (provided inside " ")
...their method selects only the kth "minimum" value from X +Y in...
It bounds the number of "corrupt" elements in the soft heap....
this can be seen as a more constrained form of "the" heap...
Due to its simplicity and contiguous memory access, it has "a" fast performance in...
used to indicate that this is the "minimum" value in the layer product..
W.l.o.g., there is "no" guaranteed ordering of the form...
The values from every element in each layer product in q "are" generated...
Each (u′, v′) matches "" exactly one layer product (u′−1, v′−1)...
Table 1: N = 4000, K = 2000...17.2 ... 0.422... 0.0409=0.0268+0.0141

Experimental design

The research question is well defined but it is a new optimal algorithm in addition to other optimal algorithms that already exist (for eg. soft heap based approach). The novelty of the paper is the usage of layer ordered heaps.

In the results section, the table is not explained in depth i.e. how different methods compare and why they are giving such results. Also, it would have been interesting to see such comparison for different values of alpha and then compare the average speed up with other methods.

I feel that the authors should provide some brief introduction to the naive algorithm stated in the results section.

Validity of the findings

I think that the manuscript lacks a concluding paragraph that wraps up everything.

Reviewer 2 ·

Basic reporting

The authors present a new algorithm for the problem of selecting the top k values from the Cartesian sum X + Y. The algorithm is simple and runs in optimal O(n + k) time by making use of Layered-Ordered Heaps. An algorithm with the same time complexity previously existed, but the authors demonstrate through experiments that the algorithm they present is faster in practice. They additionally provide an implementation of their algorithm in python.

Pros: I believe that the techniques being used here provide a very simple algorithm for solving this problem. Additionally, despite having the same asymptotic time complexity, the new algorithm appears to be significantly faster in practice.

Cons: Even though I believe that the algorithm is correct, I found many of the statements made in the proofs are hard to follow and feel they require more justification. Also, there are minor issues I take with some of the notation.

The overall structure of the proof appears to be a good way to go about proving the correctness and time complexity of the algorithm. That is, the lemmas seem to be the right ones to prove the overall result. However, the proofs of the lemmas could use additional work. I have described below the main portions of the individual proofs that could use further justification, along with some additional minor notational issues and typos.

Once these revisions are made and the proofs are satisfactorily clear, I feel that the presented algorithm is interesting enough that the work should be accepted.

Line 56: limit should be on u, not i.

Line 80: comma before X_2^(u) + Y_1^(v)

Line 83 and following paragraph: I think "false" and "true" within the tuple are unnecessary and just confused me at first. It seems 0 and 1 could be used immediately. Also, before the same line where the ordering on tuples is introduced, it should be stated that lexicographic ordering is being used. Finally, I'm not sure why (u,v) itself needs to be in the tuple at all.


Proof of Lemma 2.1 :
Line 112: "structures of pops of the form ... " Could this be made more explicit? Is there a better word than "structures", and can some justification be given? This line seems critical, after that I agree with the remaining proof.


Proof of Lemma 2.2: this is fine.


Proof of Lemma 2.3 ;
The mention of maximum and minimum tuples in the statement of the Lemma made me think that the proof was to prove all maximum tuples are in ascending order and all minimum tuples are in ascending order, but what the proof is actually trying to show that all tuples are popped in ascending order.

Line 128: "meaning that for all v >= r <= b". I am confused by this notation.
Overall, I believe this second paragraph could use much more justification. How does at "least one such floor(a,r) must be in H" go-to "all floor(a,r) must be popped before floor(u,v)"?

Similarly, I feel more justification is needed in the third paragraph. How does "at least one ..." lead to floor(a,b) being popped.


Line 141: the select notation is introduced for the first time in this way.

Line 179: Approximately less-or-equal-to should be made more rigorous.

Lines 180 and line 190, "area"? Perhaps just say product.

Line 199: "on the hull" should be "on the heap"

Section 2.4 seems unnecessary.

Very minor: I prefer sentences beginning with a word rather than mathematical notation, as was done in a few places.

Experimental design

no comment

Validity of the findings

no comment

Reviewer 3 ·

Basic reporting

In the introduction section, I would suggest the author add more practical applications for selecting the top-k problem before explaining the existing solutions to enhance the importance of the work.

Although the structure of the article is in an acceptable format, It would be better if the author provides some examples or figures for illustrations of the construction part or even the definitions to help readers understanding. For instance, in section 1.2 I would suggest adding an example including a figure to illustrate layer-order heaps and show the partitioning for some alpha.

I would suggest adding a pseudo-code including all the important steps from phases so that readers can see all the data structures (with their initializations) and the main algorithms without distracting by details and proofs.

Experimental design

no comment

Validity of the findings

no comment

Additional comments

In this paper, the author presents an optimal algorithm to output the top-k values of X_i+Y_j based on layer-ordered heaps. The solution provided in the paper is not complicated and it is easy to follow as it only uses linear time one-dimensional selection. In addition, the algorithm has fast performance since it uses data structures contiguous in memory and it is cache efficient.

This paper is well written and has been organized properly by providing appropriate sections. The problem is well stated, and the solution is compared properly with the related algorithms in the last section of the paper. The python implementations of all phases of the algorithm have been provided properly. The presented algorithm achieves 295X speed-up over the naive approach and 18X speed-up over the soft heap approach.

As far as being comprehensible, the algorithm and proofs provided are reliable and error-free. There exist sufficient proofs for the correctness of the algorithm as well as for the time and space complexities of each phase.

---

## Round 0.2 · accepted · Accept

Please prepare the manuscript for final submission taking in account any final comments that reviewers gave.

Reviewer 1 ·

Basic reporting

I think that the authors have incorporated the changes and the final version of the paper looks good and can be accepted.

Experimental design

no comment

Validity of the findings

no comment

Reviewer 2 ·

Basic reporting

I find the changes made by the author make the proofs more explicit and easier to verify. One addition that has a significant impact on the paper is the addition of Figure 1. At least for myself, it encourages visualizing this algorithm as processing the values on a grid in a particular order, rather than only contemplating the correctness of inequalities. Conceptualizing the algorithm this way makes the proofs significantly easier to understand.

The remaining comments I have are minor:

All proofs have two QED symbols.

I feel that references to figures, lemmas, phases, should be capitalized. For example, "An illustration of these phases is provided in figure 1." should be "An illustration of these phases is provided in Figure 1."

Experimental design

no comment

Validity of the findings

no comment

Reviewer 3 ·

Basic reporting

no comment

Experimental design

no comment

Validity of the findings

no comment

Additional comments

In the revised version, the authors have added an example based on my last review and have explained the corresponding algorithm. Regarding my other comment, I have been convinced that their python code is sufficient as it makes their implementation reproducible. The new version is well written and I recommend to be published.